# Supplementary stocking selects for domesticated genotypes

Ingerid J. Hagen[1], Arne J. Jensen[1], Geir H. Bolstad[1], Ola H. Diserud[1], Kjetil Hindar[1], Håvard Lo[2] & Sten Karlsson[1]

Stocking of hatchery produced fish is common practise to mitigate declines in natural populations and may have unwanted genetic consequences. Here we describe a novel phenomenon arising where broodstock used for stocking may be introgressed with farmed individuals. We test how stocking affects introgression in a wild population of Atlantic salmon (*Salmo salar*) by quantifying how the number of adult offspring recaptured in a stocked river depend on parental introgression. We found that hatchery conditions favour farmed genotypes such that introgressed broodstock produce up to four times the number of adult offspring compared to non-introgressed broodstock, leading to increased introgression in the recipient spawning population. Our results provide the first empirical evidence that stocking can unintentionally favour introgressed individuals and through selection for domesticated genotypes compromise the fitness of stocked wild populations.

[1] Norwegian Institute for Nature Research (NINA), P.O. Box 5685 Torgarden, 7485 Trondheim, Norway. [2] Norwegian Veterinary Institute, P.O. Box 5695 Torgarden, 7485 Trondheim, Norway. Correspondence and requests for materials should be addressed to I.J.H. (email: ingerid.arnesen@nina.no)

 

It has long been suspected that genetic variation resulting from domestication selection may be maintained in wild populations as an inadvertent outcome of stocking procedures that are motivated by conservation purposes[1]. Here, we show that supplementary stocking of a wild population may act contrary to its conservation goals when broodstock are introgressed with escaped farmed individuals. Our study is made possible by a unique model system that allows us to estimate reproductive success of broodstock and proportion farmed ancestry[2] in a large number of wild individuals.

Supplementary stocking of wild populations by the release of hatchery produced juveniles for conservation or harvest is being practiced worldwide for close to 180 anadromous and marine fish species[3]. Although release of hatchery produced juveniles may be important in sustaining endangered populations, there is a growing body of research suggesting negative effects of this practice, including loss of genetic variation[4], loss of adaptation[5], change of population structure[5], reduction of effective population size[6], epigenetic changes[7,8] and genetic changes from unintentional selection[9]. Here, we show that this picture is further complicated if wild individuals are introgressed with escaped farmed genotypes from aquaculture.

Artificial selection for economically important traits and genetic drift in the breeding lines of domesticated animals have shifted the allele frequencies, gene expression profiles and phenotypes away from those of their wild conspecifics[10–13], and presumably away from their selective optima in the wild. Introgression from domesticated genotypes into wild populations may, therefore, lead to negative effects in the recipient populations[14,15] and bears obvious relevance to hatchery supplementation programmes in ecosystems where conspecific domesticated farmed escapees are present. A notable example is the Atlantic salmon (*Salmo salar*), for which supplementation programmes[16] and intense aquaculture[17] overlap across the native range on both sides of the Atlantic[18,19]. Owing to 12 generations of selective breeding[20,21], farmed salmon are phenotypically[22–27] and genetically[28] different from wild salmon and heavily domesticated. It is estimated that in Norway, several hundred thousand farmed salmon escape net pens annually[29] and although escapees have a high mortality[29,30] they may in some rivers outnumber wild spawners[31–34]. Despite reduced reproductive success and survival[35–37], the high number of escapees leads to admixture between farmed and wild conspecifics and subsequent introgression from domesticated genotypes into wild populations[38,39]. Observations in a large number of wild populations show that genetic introgression from escaped farmed salmon alters important life-history traits such as age and size at maturity[40], and in situ river experiments show that farmed salmon and hybrids have lower reproductive success and survival than wild salmon[36,37,41].

Escaped farmed salmon can generally be distinguished from wild salmon by growth patterns in the scales, but individuals that have escaped at a young age can be difficult to tell apart from hatchery produced juveniles[42]. Hybrids between escaped farmed salmon and wild salmon cannot be distinguished from wild salmon using scales. Consequently, farmed escapees and hybrids have been used as broodstock in supplementation programmes[39]. Farmed salmon are selected for rapid growth and high survival in captivity[22,43], and their offspring outgrow wild salmon under hatchery conditions[23]. Introgressed broodstock are, therefore, expected to produce offspring that outcompete those of pure wild ancestry in the hatchery, but with a lower success after release[40].

An introgressed population of Atlantic salmon with a supplementary stocking programme in River Eira in Norway is a unique system for studying the combined effects of introgression from farmed genotypes and supplementary stocking on the recipient wild population. Around 50,000 hatchery-reared smolts (out-migrating juveniles) are released into the Eira annually, while about 17,000 smolts are produced naturally. On average 41% more eggs were fertilised than smolts being released, thus there is a strong potential for selection in the hatchery. Hatchery-reared individuals make up approximately 30–50% of the total spawning population[44]. Returning salmon are harvested during the summer angling and in autumn during broodstock collection. The river is situated in a region of intensive salmon farming[45] and is affected by genetic introgression from farmed escapees[39]. We used a set of genetic markers[46] and a method developed for quantifying unidirectional geneflow[2] to estimate the proportion farmed ancestry (introgression) at the individual level[40]. Our data sets comprise (1) individual estimates of proportion farmed ancestry in hatchery-reared and wild-born broodstock from seven brood years (cohorts) and the number of recaptured adults from each broodstock pair, and (2) individual estimates of proportion farmed ancestry in returning adults of wild-born and hatchery-reared origin (distinguished by fin-clipping, scale reading and parentage assignment) from 20 run years over a 30-year period. First, we estimated the number of recaptured adult offspring from broodstock pairs with varying proportions of farmed ancestry and investigated potential maternal and paternal effects due to parental environmental background (hereafter hatchery-reared or wild-born). Secondly, we estimated introgression in 20 run years of returning adults of hatchery-reared and wild-born origin.

We find that hatchery supplementation may lead to unintentional selection for genotypes associated with domestication, and subsequently increase the level of introgression in the recipient population.

## Results

**Stocking in River Eira**. To investigate the effect of parental introgression on reproductive success under hatchery conditions, we related the proportion farmed ancestry (introgression) for all reproducing broodstock pairs over seven brood years (2005–2011) to the number of offspring recaptured as adults in the River Eira. Moreover, to elucidate the underlying mechanisms driving the observed response in offspring number, we also investigated the relationships between egg production and introgression in broodstock dams as well as introgression and smolt size in adult hatchery-reared spawners. Adult spawners caught during the recreational fishery in the Eira were genetically assigned to their broodstock parents based on 81 nuclear single nucleotide polymorphisms (SNPs). From this, we identified 878 offspring belonging to 85 full sibling groups and 1–43 (mean 10.3) offspring recaptured as adults per broodstock pair (see Supplementary Table 1 for details about crossings and family groups). Among the broodstock, 55% of the dams and 65% of the sires were previously released hatchery fish. Of the 85 crosses, 7 were wild-born × wild-born, 54 were hatchery-reared × hatchery-reared and 24 were wild-born × hatchery-reared, with no bias as to whether the sire or the dam was wild-born. The proportion farmed ancestry was on average 0.303 and 0.113 in hatchery-reared and wild-born broodstock, respectively.

**Effect of introgression in broodstock**. The proportion farmed ancestry shared by the broodstock pair had a remarkably strong effect on reproductive success when the dam was wild-born: the number of offspring for a broodstock pair with 100% farmed ancestry corresponds to a factor of 5.59 (95% CI: 1.28–24.38) relative to a pair with no farmed ancestry. Controlling for the number of eggs produced by each dam improved the model by 17.08 AIC scores and caused a marginal reduction of the effect of introgression to 4.55 (95% CI: 1.23–16.80) adult offspring for

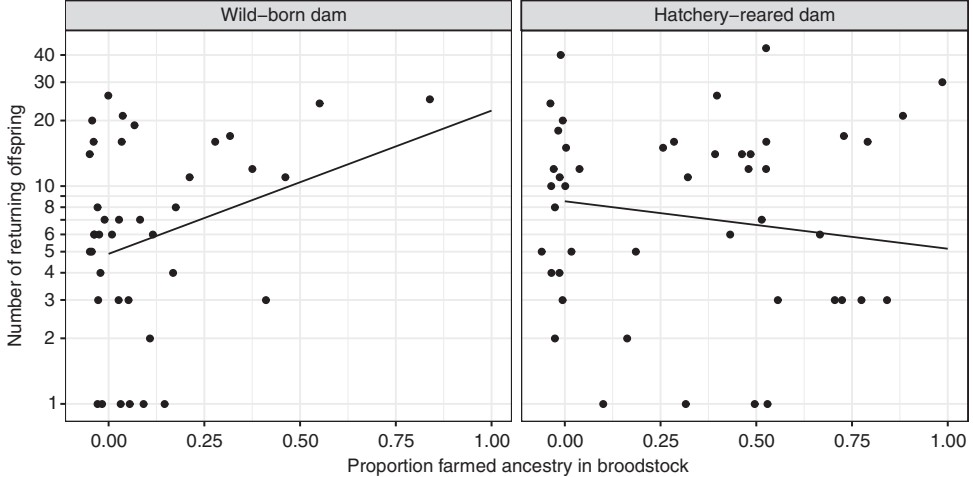

**Fig. 1** Effect of introgression on the number of recaptured adult offspring from wild-born and hatchery-reared dams. Introgressed wild-born dams produce more recaptured adult offspring than wild-born dams with no farmed ancestry. Hatchery-reared dams produce more offspring than wild-born dams but show no response to introgression. Lines represent model predictions from least square regression. See Table 1 for parameter estimates. Source data are provided as a Source Data file

broodstock pairs with 100% farmed ancestry (Fig. 1 and Table 1). No effect of broodstock introgression on the number of recaptured adult offspring was found when the dam was hatchery-reared (Fig. 1, Table 1 and Supplementary Table 2). While this lack of an effect of farmed ancestry in hatchery-reared broodstock is surprising, a possible explanation may be that a positive effect in the hatchery is counteracted after release by a larger negative effect of being second generation hatchery-reared. Multiple generations in captivity may cause cumulative negative effects on fitness components in salmonids[47]. However, the combined effects of introgression and captive rearing and how these factors affect different life-history stages of wild salmon is largely unknown. Hatchery-reared broodstock dams produced 1.75 (95% CI: 1.11–2.78) times more adult offspring than wild-born dams, while no paternal effects were observed (Table 1 and Supplementary Table 2). The increased reproductive success of hatchery-reared broodstock in supplementation programmes is expected[48] and will increase the introgression in the recipient population as hatchery-reared broodstock are more introgressed. Maternal effects influencing juvenile offspring size has been documented in salmonids[49,50] and to elucidate the underlying mechanisms for the maternal effect observed in this study, we investigated how introgression affects egg production. The effect of introgression on egg size mirrored that of reproductive success: an expected[51,52] reduction in egg size by a factor of 0.86 (95% CI: 0.80–0.93) was observed for hatchery-reared dams compared to wild-born dams. Again, a strong effect of introgression was observed in wild-born dams: individuals with 100% farmed ancestry produced eggs that were smaller by a factor of 0.67 (95% CI: 0.51–0.89), compared to wild-born dams with no farmed ancestry, while no significant response was found for hatchery-reared dams (Supplementary Fig. 1 and Supplementary Tables 3 and 4). We found no effect of egg size on number of recaptured offspring (Supplementary Table 2) thus suggesting that natural selection known to favour large eggs in nature[51] is relaxed under hatchery conditions and that the selective advantage that wild-born dams gain from producing large eggs is removed. It is apparent that introgression alters properties of the egg, and it is unlikely that these effects are limited to size alone. Interestingly, the response in reproductive success and egg size of hatchery-reared dams mirrored that of wild-born dams for which genotypes are of farmed ancestry, but with a smaller effect size. While the effect of introgression is due to 12 generations of selective

breeding, the effect of hatchery-reared dams is most likely due to epigenetic effects[7–9]. The environmental background of dams, egg number and weight of the dams (which affects egg number) influenced the number of recaptured offspring, but with smaller effect sizes than that of introgression (Table 1 and Supplementary Tables 5 and 6). Controlling for these factors did not diminish the effect of introgression, which under hatchery conditions may lead to a more than four-fold increase in reproductive success for wild-born individuals.

**Effect of introgression on growth**. To investigate whether introgression affects the size (mm) at which hatchery-reared individuals smoltify, we used a back-calculated measure of smolt size based on annual growth rings (circuli) in the scales and adult body length obtained at capture from adult spawners returning to the Eira. Fully introgressed hatchery-reared spawners were 6.2% larger as smolts (95% CI: 2.3% to 10.2%) than non-introgressed hatchery-reared individuals (Supplementary Table 7), which is in accordance with expected higher growth rate in introgressed smolts under hatchery conditions[21]. Introgressed wild-born spawners were also larger as out-migrating smolts (5%) than non-introgressed wild-born spawners, albeit with a large uncertainty (95% CI: –1% to 11.4%). Hatchery-reared individuals (mean = 232 mm) were on average much larger as out-migrating smolts than wild-born individuals (mean = 152 mm). Larger size at release increases survival at sea[53], and introgressed hatchery-reared individuals may thus be given an advantage through their size that may to some extent compensate for the negative selection pressure acting on introgressed individuals in nature[41]. Introgressed individuals are expected to spend fewer years at sea[40]. In the Eira, we found no apparent effect of introgression on sea age (Supplementary Fig. 2 and Supplementary Table 8), and a potential higher survival by spending shorter time at sea has, therefore, likely not contributed to a higher recapture rate of offspring from introgressed broodstock.

**Introgression in wild-born and hatchery-reared adults**. To investigate whether supplementation has affected introgression in the spawning population in the Eira we compared the proportion farmed ancestry in wild-born and hatchery-reared fish caught by anglers during the recreational fishery during 20 run years over a 30-year period. Altogether, this amounts to 1347 wild-born and

**Table 1 Parameter estimates for least square regression models with log number of recaptured adult offspring as response**

| Parameter | Introgression and hatchery origin of dam | Best model |
|---|---|---|
| Intercept | 1.52708 ± 0.25427 | −4.5564 ± 1.3358 |
| Dam hatchery background | 0.51936 ± 0.27144 | 0.5560 ± 0.2417 |
| Wild-born dam: Introgression | 1.72152 ± 0.75118 | 1.5145 ± 0.6667 |
| Hatchery-reared dam: Introgression | −0.04237 ± 0.47103 | −0.5018 ± 0.4319 |
| Log number of eggs | | 0.7069 ± 0.1526 |
| ΔAIC | 17.08 | 0.0 |

Each column gives the parameter estimates ± standard error for each parameter in the two models and the last row gives the difference in AIC score. The effect of hatchery background in dams gives the average difference in number of recaptured offspring to wild-born dams. Introgression is the proportion of farm ancestry in broodstock

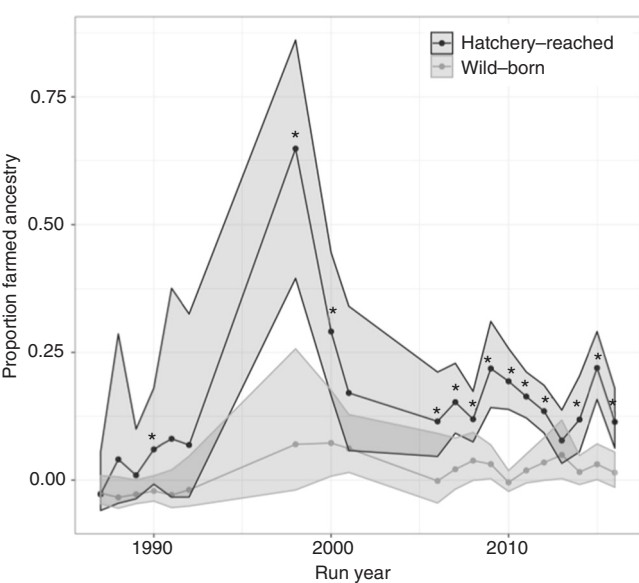

**Fig. 2** Introgression in hatchery-reared and wild-born adult spawners. Hatchery-reared adults (n = 1567) were more introgressed than wild-born adults (n = 1347) across all years compared. Dots are the estimated average for level of proportion farmed ancestry for wild-born and hatchery-reared adults, respectively. Lines connect the different run years. Shaded areas represent standard error. Differences in introgression between wild-born and hatchery-reared adults were tested using a generalised linear mixed model with a logit link. Differences that are significant at alpha level 0.05 or less are denoted with stars. See Supplementary Table 9 for details on sample sizes and significance levels. Source data are provided as a Source Data file

1567 hatchery-reared individuals. Our results show that hatchery-reared spawners have more farmed ancestry compared to wild-born spawners for nearly all analysed run years (Fig. 2), and that this difference is statistically significant for 12 out of 20 run years (Supplementary Table 9). The average level of introgression measured as proportion farmed ancestry across all run years was 0.092 in wild-born fish compared to 0.27 in hatchery-reared fish. The increased farmed ancestry in hatchery-reared adults compared to wild-born, is likely to be an effect of the higher reproductive success of introgressed broodstock. These data span three decades, and therefore suggest that increased reproductive success of introgressed broodstock (Fig. 1) is not limited to the 2005–2011 cohorts but is likely to have occurred also prior to the brood years analysed in this study.

## Discussion

While the widespread genetic introgression from escaped farmed Atlantic salmon into wild populations in Norway[39] represents a

large and unfortunate experiment of evolution, this also creates an excellent system for studying the effects of admixture between domesticated and wild conspecifics in supplemented populations. By studying the relative contribution from broodstock of farmed ancestry that is a priori known for being domesticated and adapted to hatchery conditions, we have demonstrated that domesticated genotypes can unintentionally be introduced and maintained in natural populations from supplementation programmes. In the juvenile stages, domesticated salmon have a lower survival than wild salmon under natural environments[24,36,37,54,55] but may outcompete wild salmon under hatchery conditions[23,56,57]. Our results show that the benefit hatchery-reared fish gain from having parents that are introgressed with farmed salmon can lead to a more than four-fold recapture rate for fish having parents with fully farmed ancestry than for those having fully wild ancestry and ultimately increased introgression in the recipient population (Fig. 2). This occurs despite an expected lower marine survival of farm × wild hybrids compared to wild salmon[25,36,37,41,50,58,59], and implies a strong positive selection pressure in the hatchery for individuals with a high proportion farmed ancestry. Given the difference in number of fertilised eggs and released smolts, there is a large potential for selection in the hatchery, particularly at the stage of initial feeding, when the highest mortality was observed. Selection in favour of introgressed individuals at the stage of initial feeding is expected, given that farmed[60] and hybrid[61] individuals are known to outcompete wild salmon when held in sympatry at the early life-history stage following emergence. Because introgressed hatchery-reared individuals were larger at release as 2-year smolts while mortality in the hatchery was low during the growth phase, it is likely that introgressed individuals have been favoured at two distinct life-history stages: first in the hatchery during initial feeding due to competitive behaviour[60] and faster growth[23] and then at sea, where a large size is expected to increase survival[53].

Hatchery-reared fish may in some supplemented rivers represent half or more of the total population[44,62] and domesticated genotypes are, therefore, likely to precipitate into the recipient population, even under a negative selection pressure acting on introgressed individuals[41,63] and the decreased reproductive success of hatchery-reared fish compared to wild-born conspecifics under experimental[64] and natural[47,65,66] conditions. This will inevitably put endangered populations under extra strain, many of which are supplemented because of their threatened status. From our results, a warning against the use of domesticated broodstock in supplementation programmes is warranted. This applies to ecosystems where admixture between wild and farmed conspecifics occurs[67,68], and to the use of broodstock that has been subject to unintentional domestication selection. Conservation programmes where broodstock are held in captivity for several generations for gene bank purposes[16] must be careful not to select for, and amplify, genotypes that are beneficial in captivity but maladaptive in the natural

environment. When supplementation is deemed necessary we advise that (1) the selection pressure that favours domesticated genotypes in the hatchery is reduced by creating a less artificial environment and releasing individuals at earlier life-history stages, (2) the use of hatchery-reared fish as broodstock is avoided as these are more introgressed (this study) and domesticated than wild-born fish and may accentuate unintentional domestication effects[7–9], (3) mortality is minimised and hatchery practices that sort juveniles by size to be released are avoided, as this will lead to strong selection for domesticated genotypes[23,43], and (4) introgressed individuals are identified and removed from the captive breeding population. The latter was successfully enforced in Norway in 2014, when the genetic test used in this study[2,69] became mandatory for all potential Atlantic salmon broodstock in every supplementation programme throughout the country. Such tests should be developed for all wild populations subject to supplementation in ecosystems where farmed conspecifics are present.

This study is the first to show that hatchery supplementation amplifies farmed genotypes in the offspring of hatchery broodstock and, therefore, leads to selection for genotypes associated with domestication. This effect will be accentuated in systems with genetic introgression from farmed escapees due to the high frequency of alleles that are maladaptive in the wild but beneficial in captivity. Altogether with the rapid adaptation to captivity that has been documented in other salmonids for which supplementation is frequently applied[7–9], our results add to the growing body of research that demonstrate how hatchery supplementation accentuates harmful domestication effects in the recipient population.

## Methods

**The study system**. River Eira is located in a mountainous area of western Norway at 62° 41′ N, 8° 7′ E. Its natural water discharge was 41 m³/s until reduced to 17 m³/s by three separate hydropower developments in 1953, 1962 and 1975, each removing water from the system[44]. Habitat quality in the river is compromised, and parts of the river have become unsuitable for juvenile Atlantic salmon[44]. This has led to a considerable population decline of the local salmon population, wherefore a long-term supplementation programme has been ongoing in the river for approximately five decades. The main objective for stocking of hatchery-reared fish has been to supplement the harvest opportunities and spawning population of wild salmon in the river. The reported annual catches between 1993–2015 range from 23–946 individuals caught on rod during the summer angling season[44]. To maintain the genetic integrity of the local population, only spawners caught in the river have been used as broodstock. The broodstock from the brood years 2005 to 2011 are almost without exceptions captured by seine fishing in pools known as preferred holding pools for a big proportion of the spawning population in the river. There are no indications that this fishing procedure will induce a bias in the level of introgression in the captured broodstock. In order to remove farmed escapees prior to incubation, all potential broodstock individuals have been subject to scale analysis[42]. On average 41% more eggs were fertilised than the number of smolts being released from the different brood-years, with a range of 18% (brood year 2011) to 66% (brood year 2010). The most significant mortality was observed prior to hatching and during the period of initial feeding. After the initial feeding, the mortality is reported to be zero to five individuals per month. The fish were kept at densities of 15 kg/m³ during initial feeding and 25 kg/m³ during growth. Such low densities may increase territoriality and aggressive behaviour[70]. Sick or injured individuals were removed. Feed size and change to larger pellet feed was based on the average size of fish in each holding basin. All fish were sorted and moved to larger holding basins together with conspecifics of similar size at two separate stages: during their first summer (age 0 +) and again during their second summer (age 1 +). Keeping the fish with individuals of similar size minimises competition and allows for similar growth rates. No deliberate culling of small or poor performing fish has occurred. The fish were moved to net pens at the outlet of Lake Eikesdalsvatnet—which is the source of River Eira—for acclimatisation prior to release into the river either during the second summer (1-year smolts) or third summer (2-year smolts). Prior to brood year 2009, only smolts aged 2 years were released, while from brood year 2009 and onwards both 1- and 2-year old smolts have been released. From 2009 and 2010, a relatively small proportion of the released smolts were 1-year olds (22% and 23%, respectively). In 2011 however, over half (53%) of the released individuals were 1-year olds (Supplementary Table 10).

Samples used in this study comprise (1) fish scales collected from adults captured by rod in the Eira during the summer angling season, and (2) fish scales collected from adults captured and selected as broodstock for supplementary stocking in the Eira. The samples have been assembled into two different data sets as follows: (1) broodstock spawners from brood years 2005–2011 with information on the number of offspring recaptured as adults per broodstock pair, and (2) adult fish caught by anglers during 20 run years over a 30-year period and after scale analysis being categorised as either wild-born or hatchery-reared. Individuals identified as farmed escapees based on scale analysis were removed. After brood year 1998, all hatchery-reared smolts released into the river had their adipose fin removed prior to release.

**Phenotypic measurements**. The following phenotypic measurements were available for broodstock: the wet weight (g), whether the individual was wild-born or hatchery-reared as well as the estimated number of eggs and size of eggs (ml) per broodstock dam. These measurements were chosen because large females produce more eggs than smaller individuals[71], there is a negative relationship between egg mass and number of eggs[51], domestication will lead to selection for smaller eggs[51] and epigenetic domestication effects may affect the reproductive success of broodstock[9]. Whether a broodstock individual was hatchery-reared or wild-born was determined by the presence or absence of the adipose fin and assessment of scale samples. The average egg size of each broodstock dam was estimated by counting the number of eggs needed to reach 25 cm. The number of eggs produced by each dam was estimated using the average egg size and total volume of eggs[72]. Adult spawners returning to the river were caught by rod during the summer angling season. Anglers submitted a scale sample for captured fish and reported the sex, total length (mm; from the tip of the snout to the end of the caudal fin) and presence or absence of the adipose fin (i.e., wild-born or hatchery-reared). Smolt length and annual marine growth rates were estimated by back-calculation of growth in adult scales, using the Lea-Dahl method[73].

**Assignment to brood year and wild/hatchery origin**. Prior to brood year 2005 all fish were aged and assigned as hatchery-reared or wild-born, solely by scale analyses and from records of the absence or presence of the adipose fin. From brood year 2005 and onwards we also used parentage assignment to age and assign hatchery-reared individuals. Fish with adipose fin and characterised as wild-born by scale analysis were aged according to growth patterns (annuli) in the scale[74].

**Molecular analysis**. DNA was extracted from the scale samples using DNEASY tissue kit (QIAGEN) and genotyped at 81 nuclear and 15 mitochondrial SNPs (Supplementary Table 11) using the EP1TM 96.96 Dynamic array IFCs platform (Fluidigm). Out of the nuclear SNPs, 48 have been identified as showing large genetic differences between farmed and wild salmon[46] and were used to estimate individual introgression following a STRUCTURE based method[2]. Proportion of farmed ancestry ($D$) in each individual was determined from individual estimates of the probability of belonging to farmed salmon ($P_{ind}$) by scaling to the average estimates of probability of belonging to farmed salmon in a historical reference sample of pure wild salmon from the Eira ($P_W = 0.0644$) and to reference samples of farmed salmon from all breeding lines used in Norway ($P_D = 0.903$), according to the following formulae[2]:

$$D = (P_{ind} - P_W)/(P_D - P_W)$$

Note that the average $P_D$ in Norwegian farm populations is less than one and the average $P_W$ in the historical reference sample is above zero.

**Parentage assignment**. Hatchery-reared fish were assigned to their broodstock parents by Mendelian exclusion at the 81 nuclear SNPs allowing for mismatches. In cases of one or two mismatches, we re-checked the genotypes to rule out possible genotyping errors or confirm true mismatches. All broodstock used in each brood year were set as putative parents, regardless of sex and pairs crossed. All adults assigned to the same brood year + / - 1 year were set as putative offspring, to take possible aging errors into account. The average genotype rate for offspring was 96%. Offspring with more than 20% missing genotypes were removed. Genotyping of broodstock was repeated until 100% genotype rate was achieved. Altogether, we identified 878 parent—offspring matches, out of which 26 had one mismatch and five had two mismatches. All identified parent—offspring links were verified by comparing with the documented crosses, and all mother—offspring links were verified by comparing the mitochondrial haplotype based on 15 mitochondrial SNPs.

**Statistical analysis**. Following parentage assignment, the total number of recaptured offspring per broodstock pair was recorded. In total, 85 family groups (broodstock pairs) were analysed. To analyse the effect of broodstock introgression on the number of recaptured offspring from each broodstock pair we first performed model selection on seven mixed effect models with log number of recaptured offspring as response variable and broodstock introgression (averaged between the broodstock pair), log egg number, log egg size, and environmental background of dam and sire (wild-born or hatchery-reared) as potential

explanatory variables. We also investigated whether there was an interaction between environmental background of the parent and effect of introgression. Brood year was included as a random factor (for model selection see Supplementary Table 2). The best model included the effect of environmental background of dam, log egg number and an interaction between broodstock introgression and environmental background of dam (i.e., the effect of broodstock introgression differed depending on environmental background of dam).

To analyse the effect of introgression on size and number of eggs we used a similar approach. For egg size we fitted a mixed effect model with log egg size (ml) as a response variable and introgression of dam, log size of dam (g), and environmental background of dam as potential explanatory variables (for model selection see Supplementary Table 4). Brood year was included as a random factor. The best model included environmental background of dam, log size of dam and an effect of introgression that differed depending on environmental background of dam. For number of eggs we fitted a mixed effect model with log number of eggs as response variable and introgression of dam, log size of dam (g), log egg size (ml) and environmental background of dam as potential explanatory variables (for model selection see Supplementary Table 6). Brood year was included as a random factor. The best model only included size and environmental background of dam.

To estimate the effect of introgression on smolt length, we used a mixed effect model with log back-calculated smolt length as an explanatory variable, a different intercept for each sea age and introgression of each fish as explanatory variables. Brood year was included as a random factor.

The effect of introgression on sea age (measured as probability of maturing given survival to adulthood) was analysed using the following multinomial (logit) mixed effect models:

$$\ln \frac{\Pr(y_{ijk} = 1)}{\Pr(y_{ijk} = 3+)} = a_{1i} + b_{1i}D_{ijk} + d_{1i}(D_{ij\bullet} - D_{i\bullet\bullet}) + t_{1ij},$$

$$\ln \frac{\Pr(y_{ijk} = 2)}{\Pr(y_{ijk} = 3+)} = a_{2i} + b_{2i}D_{ijk} + d_{2i}(D_{ij\bullet} - D_{i\bullet\bullet}) + t_{2ij},$$

were the subscripts $i$, $j$, and $k$, refers to sex, year of birth and individual; $a$ is the intercept, $b$ is the within year effect of proportion of domesticated genome ($D$), $d$ is the difference between the within- and among-year effect of level of introgression, and $t$ is the random effect of year. The bullet symbols denote the average taken over the indicated levels. Random effects were assumed to be independent and normally distributed on the logit scale. To evaluate the statistical support for an effect of the level of introgression on sea age for each sex, we compared the model above with one that excluded the effect of sea age (that is, a model where parameters $b_1$ and $b_2$ were set to zero for the respective sex). The same model was fitted for fish of wild and hatchery origin using the statistical software package TMB[75].

To estimate the difference in level of introgression between wild-born and hatchery-reared adult spawners we used a generalised linear mixed model with a logit link and binomially distributed residuals:

$$\log \frac{P_{\text{ind},ij}}{1 - P_{\text{ind},ij}} = a_i + b_i H_{ij} + e_{ij}$$

where the subscripts $i$ and $j$ denotes year and individuals respectively, $P_{\text{ind},ij}$ is the proportional domesticated genome for each individual, $a$ is the annual average level of introgression (logit-transformed) in wild fish, $b$ is the annual difference in level of introgression (logit) between wild-born and hatchery-reared fish, the explanatory variable $H$ takes the value 0 for wild fish and 1 for hatchery fish, and $e$ is a random effect assumed to be independent and identically normally distributed (included to account for overdispersion).

All 95% confidence intervals were estimated using the relevant estimates and their standard error multiplied by 1.96 and then transformed to the appropriate scale.

**Code availability**. R-code for the statistical models is available in Dryad Digital Repository with the identifier https://doi.org/10.5061/dryad.1nh877d. R is freely distributed at https://cran.r-project.org/.

**Reporting summary**. Further information on experimental design is available in the Nature Research Reporting Summary linked to this article.

## Data availability

The data supporting the findings of this study are available in the Dryad Digital Repository with the identifier https://doi.org/10.5061/dryad.1nh877d. The source data has been uploaded on Dryad as a SourceData.xlsx file. All other relevant data is available upon request.

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

## Acknowledgements

We thank Ola Ugedal at the Norwegian Institute for Nature Research (NINA) for valuable comments and discussions, Line Birkeland Eriksen, Merethe Hagen Spets and Torveig Balstad at NINA for the genotyping, Bjørn Bjøru, Espen Holthe and Bjørn Florø-Larsen at the Norwegian Veterinary Institute for providing scale samples and data from the broodstock, and Monika Klungervik at Statkraft AS for providing information of broodstock crossing, and Thomas Moen at AquaGen AS for providing the script for parentage analysis. The study was supported by funding from Statkraft AS, the Research Council of Norway (grant no. 254852 and 275862) and from NINA.

## Author contributions

S.K., A.J.J., H.L., G.H.B. and I.J.H. designed the study. G.H.B. and I.J.H. conducted the statistical analysis and O.H.D. provided statistical advice. A.J.J. and H.L. provided tissue samples and scale reading data from the salmon caught in recreational fishing and from broodstock, respectively. I.J.H., S.K. and K.H. wrote the Article with input from all authors. S.K. is the senior author and has been the project leader.

## Additional information

**Competing interests:** The authors declare no competing interests.

