## [Peer Review File · Nature Communications]

Reviewers' comments:

Reviewer #1 (Remarks to the Author):

This is an interesting study, which provides novel results showing important interactions between introgression from domesticated fish into wild populations, and subsequent positive selection for descendants of domesticated fish when the population is subject to supportive breeding. In other words, genetic variation and traits that have been subject to domestication selection in captivity may be maintained in wild populations as an inadvertent outcome of procedures that were in fact instigated for conservation purposes. It has been suspected for some time that such dynamics could occur, but to my knowledge this is the first time it has been demonstrated.

I think the idea behind the study is really good and the time scale it spans is impressive. Of course, one can always criticize that this represents a study from just one river and the outcome might be different in other environments/settings. However, this would not be fair given the time scale covered in the present study and the efforts that have gone into it. In fact, although the study is a bit weakened by selection favouring individuals with high farmed admixture proportions in one time period, but not in the other time period, this could also be turned into a strength, as it also serves as a sort of biological replicate. Perhaps, the authors could make more use of this and discuss how general the findings are expected to be and which factors could potentially change the outcome in other systems.

As a whole the study is interesting and the results are clearly important in a conservation context and deserve wide-spread attention. From that perspective, the paper should definitely be published. The question is then if it is of sufficient quality for Nature Communications. When I first read through the paper I had some concerns about especially the low number of genetic markers that conclusions are based on and the adequacy of some of the statistics. However, during my second reading I ended up concluding that the findings are robust and uncertainties generally well accounted for. Hence, I do think the quality of the science also fulfils the requirements for publication in a high-quality journal.

Below I provide a number of comments, which mostly concern clarification of specific points.

1. L. 12 and elsewhere. I think you should avoid the term "genomes" unless you specifically refer to processes occurring in the genome or specific genes or chromosome regions. The Atlantic salmon genome is > 3Gbp and you analyze only 81 nuclear SNPs. When I first read "genomes" I got the impression that you had conducted some sort of genome scan and identified chromosomal regions under selection. This is clearly not the case. You have markers that allow you to estimate individual admixture proportions, but you cannot say anything about processes at the genomic level. Readers will probably be disappointed when they (like me) expect to read about findings at the genomic level.

2. L. 18 and elsewhere. "...domesticated traits...". Please be specific and mention the traits. From the following text I guess this may be difficult, perhaps with the exception of size of smolts. Alternatively, you could write "...to unintentional selection favouring individuals of domesticated ancestry...".

3. L. 38. Again, which domesticated traits?

4. L. 72. It would be good if you already here mentioned how you distinguish a fish "born" in a hatchery from a naturally produced spawner (that is, by fin-clipping).

5. L. 81. Please state here what the genotyping consisted of.

6. L. 85-86. This sentence is difficult to follow. I guess it comes down to understanding how the same individuals can belong to several families. Please clarify.

7. L. 101. What is "maternal mass"? Weight?

8. L. 147-150. You state that the hatchery fish consistently have higher admixture proportions of domesticated fish relative to naturally reproduced salmon. I wonder if there could be a simple explanation for that, e.g. that the procedures for collection of spawners could favour inclusion of fish of domesticated origin? For instance, there is evidence for different run and spawning time for domesticated and wild salmon. Overall, the procedures for sampling and choosing individuals for hatchery-rearing should be better explained.

9. L. 168-171. Do mortality rates in the hatchery suggest that such strong selection could occur?

10. L. 256. How were SNPs genotyped, that is which platform?

11. L. 257. What were the mtDNA SNPs used for?

12. L. 267-274. I think your procedure for estimating admixture is valid, but I am really puzzled why you did not just use e.g. STRUCTURE, NewHybrids or similar software, the robustness of which are well-documented.

13. L. 266-269. This point is quite similar to the preceding. Why did you not simply use a method like e.g. Cervus for parentage assignment? Although the method you use seems valid, I do not really see the point for doing it in this way. Did you use specific software or scripts for this?

14. L. 272. 20% missing data as a threshold sounds like quite a lot. Given that you use a rather low number of SNPs (81) can you then say something about confidence by which individuals with 15-20% missing data can be assigned?

Reviewer #2 (Remarks to the Author):

Nature Communications review of 'Supplementary stocking selects for domesticated genotypes'.

Main comments

This study investigates the genetic long term effects of supplementing wild populations with hatchery reared individuals. The species investigated is the Atlantic salmon and the data is collected across a multiyear study from a river system in Norway. Specifically, the study seeks to quantify the extent to which domesticated genomes are preferentially selected compared to wild genomes by analysing broodstock fish and resulting offspring that return to the river following release as adults. The study finds that the broodstock contains many parents with introgressed genomes and detect that offspring survival is also greater in those fish that have a larger proportion of their wild genome introgressed with the domesticated genome.

While this study is intriguing, I am left wondering what the main message of the studies findings may be. Is it that scientists involved in restocking programmes need to better take the genetic composition of their broodstock fish into account to avoid unintentional selection? If so this needs to be made much clearer (but also somewhat common knowledge). One aspect that is blatantly missing in the study is knowledge of the genome composition of smolts at release. This would allow one to estimate selection in the wild against introgressed vs pure wild genome (based on what was released vs which

ones returned). I guess there is not much that can be done about this though.

This also brings me to a related issue. The causes for the high survival of the domesticated salmon is unclear, and are little discussed. The aquaculture raising protocols are little elaborated on but I wonder if size grading, something that is routinely used in salmon aquaculture, was applied during larval rearing? Given that domesticated salmon grow much faster than wild salmon, this would likely have led to a much higher output of partially domesticated smolts, resulting in the higher return of salmon with highly introgressed genomes (even if their survival would be somewhat lower compared to wild salmon). Under such a scenario I am again left wondering what we have learned from this study? That aquaculture size selection (grading) preferentially selects for domesticated phenotypes and that this can have severe effects on the genetic composition of returning adults? This is not really unintentional selection, but intentional selection to remove small growing fish, with the consequence that domesticated genomes are also preferred, as a side effect. While I am finding that hence the real take away message is still quite unclear, I also find that a lot of the novel results are common knowledge, in that the impact of restocking practices have been long now discussed under this light, and the negative consequences have been well articulated. Having said that, this has never been done with such a powerful dataset as the one presented in this study. I should highlight that this paper is based on a long term data set that is quite unique and thus of relevance in this context as it allows the -often only verbal models- to be tested with real data. This is of course, of immense value and I do not mean to undermine this.

Minor comments

I have a number of minor issues with the study. The language is a little poor and typos are throughout the paper. This needs to be fixed.

Some of the results are a little overstated, see abstract line 18...'stocking leads to unintentional selection for domesticated traits' . I would urge the authors to write 'stocking can lead...'

I also wonder about the statement in the intro, line 38-39 'introgression of domesticated traits into wild populations therefore leads to negative effects for the recipient populations' . Is that really always the case? I think this is somewhat simplistic. Faster growing salmon can maybe outcompete wild ones? In your study you find more introgressed individuals return to the hatchery....we do not know the starting numbers, but this may indicate that they are not doing so poorly.

Page 13 line 256. The authors refer to the 81 SNPs used and describe that the selection was based on ref 43. But this ref only described 60 SNPs

Reviewer #3 (Remarks to the Author):

Hagen et al. evaluated reproductive success in and genetic interaction between hatchery and wild origin fish, using genetic data from Atlantic salmon. Based on the results, the authors claim that they found evidence of pronounced introgression of genes selected for domesticated traits in wild populations.

General comments

The research subject is very important in conservation biology, and the study species is of central interest in the North Atlantic. However, I am concerned about its readability, which makes me difficult to evaluate its scientific validity and novelty. Below I provide a few examples.

The authors mentioned 85 family groups (Line 85) and 872 parent-offspring matches (273). I suppose that 872 contains both mother-offspring and father-offspring matches. Does it mean that on average each family had c.a. five offspring ($872/2/85=5.13$ offspring/family)? It means the population is growing rather rapidly, despite the high level of genetic introgression. Population dynamics could be

very relevant to the reproductive success, although I found no description on the population dynamics during the study period.

Adult return rates were estimated by combining data on rod catches, broodstock data, and snorkeling surveys (Line 224). Any risk of potential bias due to the different source of data in this study? For example, what if fish with more domesticated background tend to be easily caught by rod or spotted? Does it potentially change the main conclusion of this manuscript? More careful evaluation and detailed discussion would be needed before jumping into a conclusion.

This article can be considered as a follow-up study of their previous paper reporting the phenotypic effect of introgression (Bolstad et al. 2017). However, I do not see any description that tried to relate the reproductive success with their phenotypes as potential mechanism of the reported phenomenon.

Detailed comments

Abstract: no period in Line 16. More importantly, the study species should be declared earlier than the end of the abstract (Line 19).

Line 40: 'escapees' is an inappropriate wording for supplementation programs because they are meant to be released in the wild.

Line 227: Is Table S2 the same as the Supplementary Table 2 (GLMM parameters)?

Line 239: How the phenotypic measurements, other than the fish origins, were used in this study is unclear.

Line 263: How were Pw and Pd determined, and what do they mean? For example, does "domesticated salmon" in this river have on average 90% of "domesticated genes" and 10% of "wild genes"? Detailed explanation is needed.

Reviewers' comments:

Reviewer #1 (Remarks to the Author):

This is an interesting study, which provides novel results showing important interactions between introgression from domesticated fish into wild populations, and subsequent positive selection for descendants of domesticated fish when the population is subject to supportive breeding. In other words, genetic variation and traits that have been subject to domestication selection in captivity may be maintained in wild populations as an inadvertent outcome of procedures that were in fact instigated for conservation purposes. It has been suspected for some time that such dynamics could occur, but to my knowledge this is the first time it has been demonstrated.

I think the idea behind the study is really good and the time scale it spans is impressive. Of course, one can always criticize that this represents a study from just one river and the outcome might be different in other environments/settings. However, this would not be fair given the time scale covered in the present study and the efforts that have gone into it. In fact, although the study is a bit weakened by selection favouring individuals with high farmed admixture proportions in one time period, but not in the other time period, this could also be turned into a strength, as it also serves as a sort of biological replicate. Perhaps, the authors could make more use of this and discuss how general the findings are expected to be and which factors could potentially change the outcome in other systems.

In response to the comments of reviewer 2, who suggested that we investigate the underlying mechanisms for the higher survival of introgressed individuals, we approached the managers of the Eira stocking programme and requested historical records of number of eggs from each female, egg size and date of incubation. Including egg parameters in the analyses required a different statistical approach and facilitated more accurate analyses of effects relating to the broodstock female. The new analyses uncovered a much stronger response in reproductive success of introgression in broodstock dams that are themselves born in the wild (wild-born) compared to those that are hatchery-reared. There are still indications that the release of one-year smolts induced a different selection pressure during the 2009-2011 cohorts that again altered the response to introgression. However, our data is not large enough to separate the treatments into both origin of dam (hatchery-reared or wild-born) and period (early or late). A separation of the data into hatchery-reared and wild-born broodstock dams not only shows a strong effect of introgression but also sheds light on potential maternal and epigenetic effects. These results complement the findings by, i.e. Christie 2016; Nature Communications and Le Luyer 2017; PNAS, and we have chosen to pursue the hatchery/wild approach rather than early/late period. Importantly, this does not change the novel conclusions of our study, which are still that i) broodstock with farmed ancestry produce more adult offspring than broodstock of completely wild ancestry, and ii) this has led to increased introgression in the recipient population.

As a whole the study is interesting and the results are clearly important in a conservation context and deserve wide-spread attention. From that perspective, the paper should definitely be published. The question is then if it is of sufficient quality for Nature Communications. When I first read through the paper I had some concerns about especially the low number of genetic markers that conclusions are based on and the adequacy of some of the statistics. However, during my second reading I ended up concluding that the findings are robust and uncertainties generally well accounted for. Hence, I do think the quality of the science also fulfils the requirements for publication in a high-quality journal.

Below I provide a number of comments, which mostly concern clarification of specific points.

1. L. 12 and elsewhere. I think you should avoid the term "genomes" unless you specifically refer to processes occurring in the genome or specific genes or chromosome regions. The Atlantic salmon

genome is > 3Gbp and you analyze only 81 nuclear SNPs. When I first read "genomes" I got the impression that you had conducted some sort of genome scan and identified chromosomal regions under selection. This is clearly not the case. You have markers that allow you to estimate individual admixture proportions, but you cannot say anything about processes at the genomic level. Readers will probably be disappointed when they (like me) expect to read about findings at the genomic level. ***We agree with the reviewer and we have changed the term 'proportion domesticated genome' to 'proportion farmed ancestry'.***

2. L. 18 and elsewhere. "...domesticated traits...". Please be specific and mention the traits. From the following text I guess this may be difficult, perhaps with the exception of size of smolts. Alternatively, you could write "...to unintentional selection favouring individuals of domesticated ancestry...". ***We agree, and we thank the reviewer for the suggestion. We have altered the last sentence in the Abstract to the following: "Our results provide the first empirical evidence that stocking can unintentionally favour introgressed individuals and through selection for domesticated genotypes compromise the fitness of stocked wild populations."***

3. L. 38. Again, which domesticated traits?
We have changed the term 'traits' to 'genotypes'.

4. L. 72. It would be good if you already here mentioned how you distinguish a fish "born" in a hatchery from a naturally produced spawner (that is, by fin-clipping). ***We have included the requested information and altered the text as follows (Lines 74-79): "Our datasets comprise 1) individual estimates of proportion farmed ancestry in hatchery-reared and wild-born broodstock from seven brood years (cohorts) and the number of recaptured adults from each broodstock pair, and 2) individual estimates of proportion farmed ancestry in returning adults of wild-born and hatchery-reared origin (distinguished by fin-clipping, scale reading and parentage assignment) from 20 run years over a 30-year period."***

5. L. 81. Please state here what the genotyping consisted of.
We have altered the text as follows (Lines 91-92): "Adult spawners caught during the recreational fishery in the Eira were genetically assigned to their broodstock parents based on 81 nuclear SNPs."

6. L. 85-86. This sentence is difficult to follow. I guess it comes down to understanding how the same individuals can belong to several families. Please clarify.
We agree with the reviewer that the original sentence was unclear. Information about the broodstock individuals used in one or two crossings has been moved to Supplementary Table 1. We have re-phrased the text as follows (Lines 91-99): "Adult spawners caught during the recreational fishery in the Eira were genetically assigned to their broodstock parents based on 81 nuclear SNPs. From this, we identified 878 offspring belonging to 85 full sibling groups and 1 – 43 (mean 10.3) offspring recaptured as adults per broodstock pair (see Supplementary Table 1 for details about crossings and family groups). Among the broodstock, 55% of the dams and 65% of the sires were previously released hatchery fish (hereafter hatchery-reared). Of the 85 crosses, 7 were wild-born × wild-born, 54 were hatchery-reared × hatchery-reared and 24 were wild-born × hatchery-reared, with no bias as to whether the sire or the dam was wild-born. The proportion farmed ancestry was on average 0.303 and 0.113 in hatchery-reared and wild-born broodstock, respectively."

7. L. 101. What is "maternal mass"? Weight?
We agree with the reviewer and we have changed the term 'mass' to 'weight'.

8. L. 147-150. You state that the hatchery fish consistently have higher admixture proportions of domesticated fish relative to naturally reproduced salmon. I wonder if there could be a simple

explanation for that, e.g. that the procedures for collection of spawners could favour inclusion of fish of domesticated origin? For instance, there is evidence for different run and spawning time for domesticated and wild salmon. Overall, the procedures for sampling and choosing individuals for hatchery-rearing should be better explained.

We agree with the reviewer that more information should be provided with respect to the capture of broodstock. Also, we understand the reviewer's concern with respect to a potential bias in the level of introgression in captured broodstock. In the River Eira supplementation programme, farmed escapees are removed from the broodstock population based on scale analysis (reference 42). The introgression in the broodstock is therefore a result of wild born hybrids being used as broodstock and the re-use of previously released hatchery-reared individuals - which we have shown leads to higher level of introgression in the stocked proportion of the population due to selection for genotypes associated with domestication. Moreover, in our study we have shown that when hatchery-reared and wild-born broodstock are compared under the same conditions, then those of farmed ancestry produce more offspring. The broodstock are captured by seine fishing in pools that hold large proportions of the spawning population on two (sometimes one) time points each season. It is unlikely that this method should favour introgressed individuals. Indeed, an additional analysis comparing the broodstock used in the different brood years with the spawning population in the river the same year (run year) shows that (apart from brood year 2010) there is no significant difference in proportion farmed ancestry between the broodstock and the overall spawning population for the different years (see Figure 1 below – not included in the manuscript). No wild-born spawners from run year 2005 were genotyped. The high introgression in 2010 is most likely due to incidental use of several highly introgressed broodstock individuals and does not affect the main result in the manuscript. To clarify this further we have added the following information to the Methods section (Lines 244-249): "The broodstock from the brood years 2005 to 2011 are almost without exceptions captured by seine fishing in pools known as preferred holding pools for a big proportion of the spawning population in the river. There are no indications that this fishing procedure will induce a bias in the level of introgression in the captured broodstock. In order to remove farmed escapees prior to incubation, all potential broodstock individuals have been subject to scale analysis⁴²."

Figure 1: Proportion farmed ancestry in broodstock and wild spawners for run years 2006 – 2011.

9. L. 168-171. Do mortality rates in the hatchery suggest that such strong selection could occur? ***On average 41 % more eggs were fertilized than smolts being released from the different brood-years, with a range of 18 % (brood-year 2011) to 66 % (brood-year 2010). This shows that there has been a potential for selection in the hatchery, likely favouring domesticated genotypes. This mortality from eggs to smolts is higher than the 5% to 10% achieved by best practice by the national genebank program (The Norwegian Veterinary institute – National genebank program, unpublished data) for re-stocking of depleted Norwegian salmon populations. In any stocking or genebank program, some mortality is to be expected, however there exist very little published data on the average mortality in different hatcheries. In response to the reviewer’s question; there is a relatively high mortality in the Eira supplementation program. In this study we present results that indicate how a high mortality will lead to a strong selection pressure and subsequently accentuate the negative domestication effects of stocking.***

We agree with the reviewer that this information should be presented in the manuscript and we have re-phrased the introduction as follows (Lines 65-79): “Around 50 000 hatchery-reared smolts (out-migrating juveniles) are released into the Eira annually, while about 17 000 smolts are produced naturally. On average 41% more eggs were fertilized than smolts being released, thus there is a strong potential for selection in the hatchery. Individuals of hatchery origin make up approximately 30 – 50% of the total spawning population⁴⁴.” More details are presented in the Methods section (Lines 250-251): “On average 41% more eggs were fertilized than the number of smolts being released from the different brood-years, with a range of 18% (brood-year 2011) to 66% (brood-year 2010).”

Also, in response to the reviewer’s question about the Lines 168-171 (old line number) in the Discussion, we have made the following changes to the text (now Lines 189-298):

“This occurs despite an expected lower marine survival of farm × wild hybrids compared to wild salmon^{25, 36, 37, 41, 49, 57, 58}, and implies a strong positive selection pressure in the hatchery for individuals with a high proportion farmed ancestry. Specifically, introgressed hatchery-reared individuals were larger at release as two-year smolts and implies size dependent mortality where large individuals are favoured both in the hatchery and at sea⁵². Given the difference in number of fertilised eggs and released smolts, there is a strong potential for selection in the hatchery. How this mortality compares with that of other supplementation programs is not known. Size dependent mortality is unlikely to be the only mechanism involved, as farmed salmon and hybrids have a higher tolerance for crowding and stress in addition to increased growth rate²³.”

10. L. 256. How were SNPs genotyped, that is which platform?

We have included this information in the Methods section as follows (Lines 292-294): “DNA was extracted from the scale samples using DNEASY tissue kit (QIAGEN) and genotyped at 81 nuclear and 15 mitochondrial SNPs (Supplementary Table 11) using the EP1TM 96.96 Dynamic array IFCs platform (Fluidigm).”

11. L. 257. What were the mtDNA SNPs used for?

The mitochondrial SNPs were used to compare mitochondrial haplotypes between mother – offspring links identified using nuclear SNPs. This served as a verification to ensure that mother offspring links were correct. In the data from the Eira, 15 different mitochondrial haplotypes are identified. The use of the mitochondrial SNPs is described in Lines 316-318: “All identified parent – offspring links were verified by comparing with the documented crosses, and all mother – offspring links were verified by comparing the mitochondrial haplotype based on 15 mitochondrial SNPs.”

12. L. 267-274. I think your procedure for estimating admixture is valid, but I am really puzzled why you did not just use e.g. STRUCTURE, NewHybrids or similar software, the robustness of which are well-documented.

We understand the reviewers concern. The method implemented in our study is indeed based on STRUCTURE (here from Karlsson et al 2014, Ecology and Evolution, reference 2: “In STRUCTURE, individuals are probabilistically assigned to an a priori assumed number of populations based on their multi-locus genotype, so that deviations from Hardy–Weinberg and linkage equilibrium are minimized. To avoid biased results in STRUCTURE, as explained above, we analyzed one by one individual together with the center points, instead of all individuals of interest collectively, assuming two populations ($K = 2$). From the genetic assignment of reference individuals to the center points, we obtained a probability of belonging to the wild center point $P(\text{wild})$ for each individual. The corresponding probability of belonging to the farm center point is $1 - P(\text{wild})$.” In response to the above comment, we have changed the sentence to the following (Lines 294-397): “Out of the nuclear SNPs, 48 have been identified as showing large genetic differences between farmed and wild salmon⁴⁶ and were used to estimate individual introgression following a STRUCTURE based method²”.

13. L. 266-269. This point is quite similar to the preceding. Why did you not simply use a method like e.g. Cervus for parentage assignment? Although the method you use seems valid, I do not really see the point for doing it in this way. Did you use specific software or scripts for this?

Because of the low number of possible parents, an already developed script and straightforward work flow, we chose to conduct the parentage assignment by Mendelian exclusion.

14. L. 272. 20% missing data as a threshold sounds like quite a lot. Given that you use a rather low number of SNPs (81) can you then say something about confidence by which individuals with 15-20% missing data can be assigned?

We appreciate the reviewer’s concern. The average genotype rate for offspring was 96%, whilst 100% genotype rate was achieved for all broodstock. We have included average genotype rate for

offspring in the manuscript as follows (Lines 313-314): “The average genotype rate for offspring was 96% and offspring with more than 20% missing genotypes were removed.”

Moreover, we have confidence in the matches we have identified due to the following:

1) All parent-offspring links were confirmed by checking against recorded crosses.

2) All mother- offspring links were confirmed by comparing the mitochondrial haplotype based on 15 mitochondrial SNPs, which must be identical in mother and offspring. In this system there are 15 mitochondrial haplotypes and as such a high degree of genetic variation.

Reviewer #2 (Remarks to the Author):

Nature Communications review of 'Supplementary stocking selects for domesticated genotypes'.

Main comments

This study investigates the genetic long term effects of supplementing wild populations with hatchery reared individuals. The species investigated is the Atlantic salmon and the data is collected across a multiyear study from a river system in Norway. Specifically, the study seeks to quantify the extent to which domesticated genomes are preferentially selected compared to wild genomes by analysing broodstock fish and resulting offspring that return to the river following release as adults. The study finds that the broodstock contains many parents with introgressed genomes and detect that offspring survival is also greater in those fish that have a larger proportion of their wild genome introgressed with the domesticated genome.

While this study is intriguing, I am left wondering what the main message of the studies findings may be. Is it that scientists involved in restocking programmes need to better take the genetic composition of their broodstock fish into account to avoid unintentional selection? If so this needs to be made much clearer (but also somewhat common knowledge).

In the discussion we give four specific advices for how unintentional selection in the hatchery for domesticated genotypes can be avoided. We have also added a sentence in the discussion stating the following (Lines 180-184): "By studying the relative contribution from broodstock of farmed ancestry that is a priori known for being domesticated and adapted to hatchery conditions, we have demonstrated that domesticated genotypes can unintentionally be introduced and maintained in natural populations from supplementation programs". This is followed by a warning (Lines 205-208) about the using of broodstock that have domesticated genotypes, either because they have farmed ancestry or because they for other reasons have been subject to domestication selection.

One aspect that is blatantly missing in the study is knowledge of the genome composition of smolts at release. This would allow one to estimate selection in the wild against introgressed vs pure wild genome (based on what was released vs which ones returned). I guess there is not much that can be done about this though.

This also brings me to a related issue. The causes for the high survival of the domesticated salmon is unclear, and are little discussed. The aquaculture raising protocols are little elaborated on but I wonder if size grading, something that is routinely used in salmon aquaculture, was applied during larval rearing? Given that domesticated salmon grow much faster than wild salmon, this would likely have led to a much higher output of partially domesticated smolts, resulting in the higher return of salmon with highly introgressed genomes (even if their survival would be somewhat lower compared to wild salmon). Under such a scenario I am again left wondering what we have learned from this study? That aquaculture size selection (grading) preferentially selects for domesticated phenotypes and that this can have severe effects on the genetic composition of returning adults? This is not really unintentional selection, but intentional selection to removes small growing fish, with the consequence that domesticated genomes are also preferred, as a side effect. While I am finding that hence the real take away message is still quite unclear, I also find that a lot of the novel results are common knowledge, in that the impact of restocking practices have been long now discussed under this light, and the negative consequences have been well articulated.

Having said that, this has never been done with such a powerful dataset as the one presented in this study. I should highlight that this paper is based on a long term data set that is quite unique and thus of relevance in this context as it allows the -often only verbal models- to be tested with real data. This is of course, of immense value and I do not mean to undermine this.

As the reviewer points out, despite negative domestication effects from stocking being known for years, conclusive evidence is rare. This study is the first to document such an effect in a wild population subject to stocking. Importantly, our study documents a difference in number of recaptured adults from broodstock pairs of high and low level of introgression and subsequently shows an effect on the spawning population as a whole. Introgressed individuals are selected against in nature. The response we have measured is based on individuals that have spent 1-4 years at sea, where they have been subject to a number of stochastic events and – importantly – a negative selection pressure on introgressed individuals. The fact that we can still document a difference in returning adult offspring from broodstock of varying degrees of introgression indicates that natural selection is not able to counteract the negative domestication effects of stocking, particularly when there is a high degree of admixture with farmed escapees.

However, we agree with the reviewer that a weakness of our study is the lack of genetic information on the released smolt. Hence, we can only estimate net selection for domestication, and not separate the selection in captivity and in the wild. Following the reviewer's suggestion, we have taken effort to come up with a clearer message in this new version of the manuscript. Specifically, we have altered the introduction to more clearly state the motivation of the study. The text now reads (Lines 19-21): "It has for some time been suspected that genetic variation resulting from domestication selection may be maintained in wild populations as an inadvertent outcome of stocking procedures that are motivated by conservation purposes¹. Here we show that supplementary stocking of a wild population may act contrary to its conservation goals when broodstock are introgressed with escaped farmed individuals. Our study is made possible by a unique model system that allows us to estimate reproductive success of broodstock and proportional farmed ancestry² in a large number of wild individuals."

With respect to the reviewer's reflections around size grading, we have made efforts to find more information about hatchery practises in the historical records of the Eira supplementation programme and use the relevant information in the analyses. No intentional culling of fish has occurred in the hatchery (personal communication with managers of Eira stocking program). Additional factors that we considered important are the number of egg from each female and egg size. Adding this information to the analyses has shed light on the underlying mechanisms for the increased number of offspring produced by introgressed broodstock. Also, we have used data on back-calculated smolt size and related individual levels of introgression to estimated smolt size. These results are discussed in a new section of the Discussion (Lines 192-198): "Specifically, introgressed hatchery-reared individuals were larger at release as two-year smolts and implies size dependent mortality where large individuals are favoured both in the hatchery and at sea⁵². Given the difference in number of fertilised eggs and released smolts, there is a strong potential for selection in the hatchery. How this mortality compares with that of other supplementation programs is not known. Size dependent mortality is unlikely to be the only mechanism involved, as farmed salmon and hybrids have a higher tolerance for crowding and stress in addition to increased growth rate²³."

Minor comments

I have a number of minor issues with the study. The language is a little poor and typos are throughout the paper. This needs to be fixed.

Some of the results are a little overstated, see abstract line 18...'stocking leads to unintentional selection for domesticated traits' . I would urge the authors to write 'stocking can lead...'

I also wonder about the statement in the intro, line 38-39 'introgression of domesticated traits into wild populations therefore leads to negative effects for the recipient populations' . Is that really always the case? I think this is somewhat simplistic. Faster growing salmon can maybe outcompete wild ones? In your study you find more introgressed individuals return to the hatchery....we do not know the starting numbers, but this may indicate that they are not doing so poorly.

Following the reviewer's concerns, we have made efforts to improve the language in the manuscript and corrected typos. In accordance with the suggestion from reviewer 1 and 2, Lines 16 - 18 in the abstract has been altered to "Our results provide the first empirical evidence that stocking can unintentionally favour introgressed individuals and through selection for domesticated genotypes compromise the fitness of supplemented wild populations." Moreover, Lines 38-39 (now 38-41) have been altered to the following: "Introgression from domesticated genotypes into wild populations may therefore lead to negative effects in the recipient populations^{14, 15} and bears obvious relevance to hatchery supplementation programs in ecosystems where conspecific domesticated farmed escapees are present." With respect to the reviewer's reflections pertaining to the survival and reproduction of introgressed salmon, it is possible that the large size of introgressed hatchery individuals partly compensate for a negative selection pressure at sea. However, we should at least expect introgressed individuals to do worse than fish of pure wild origin as shown by Hindar et al. (2006), and references therein.

Page 13 line 256. The authors refer to the 81 SNPs used and describe that the selection was based on ref 43. But this ref only described 60 SNPs.

We thank the reviewer for pointing out this discrepancy. The 81 nuclear SNPs are among the 200 that were tested in the above-mentioned reference. Because not all 81 are listed in the supplementary table in reference, we have chosen to include a full list of all 81 SNPs and the 15 mitochondrial SNPs. The list is found in Supplementary Table 11 and contains SNP ID, NCBI reference and whether the SNP is used for P(wild) assessment or only parentage assignment. Concordantly, we have altered the sentence on lines 292-294 to the following: "DNA was extracted from the scale samples using DNEASY tissue kit (QIAGEN) and genotyped at 81 nuclear and 15 mitochondrial SNPs (Supplementary Table 11) using the EP1TM 96.96 Dynamic array IFCs platform (Fluidigm)."

Reviewer #3 (Remarks to the Author):

Hagen et al. evaluated reproductive success in and genetic interaction between hatchery and wild origin fish, using genetic data from Atlantic salmon. Based on the results, the authors claim that they found evidence of pronounced introgression of genes selected for domesticated traits in wild populations.

General comments

The research subject is very important in conservation biology, and the study species is of central interest in the North Atlantic. However, I am concerned about its readability, which makes me difficult to evaluate its scientific validity and novelty. Below I provide a few examples.

The authors mentioned 85 family groups (Line 85) and 872 parent-offspring matches (273). I suppose that 872 contains both mother-offspring and father-offspring matches. Does it mean that on average each family had c.a. five offspring ($872/2/85=5.13$ offspring/family)? It means the population is growing rather rapidly, despite the high level of genetic introgression. Population dynamics could be very relevant to the reproductive success, although I found no description on the population dynamics during the study period.

The population is harvested. This is mentioned in Lines 240-242 as follows: "The main objective for stocking of hatchery-reared fish has been to supplement the harvest opportunities and spawning population of wild salmon in the river". We appreciate that the dynamics concerning harvest should be clarified further. We have attempted to do so by adding the following information (Line 242-243): "The reported annual catches between 1993 – 2015 range from 23 – 946 individuals caught on rod during the summer angling season⁴⁴."

Adult return rates were estimated by combining data on rod catches, broodstock data, and snorkeling surveys (Line 224). Any risk of potential bias due to the different source of data in this study? For example, what if fish with more domesticated background tend to be easily caught by rod or spotted? Does it potentially change the main conclusion of this manuscript? More careful evaluation and detailed discussion would be needed before jumping into a conclusion.

Snorkelling surveys were used to estimate overall return rates for released smolts of different cohorts (presented in Supplementary Table 10). We acknowledge that this can easily be confused with the number of recaptured adults per broodstock pair. We have therefore moved the information pertaining to overall return rates for the stocking programme to the table legend for Supplementary Table 10. Also, we have clearly stated that individuals identified as escaped farmed salmon were removed from the analysis. The text in the revised manuscript now reads (Lines 259-264): "The samples have been assembled into two different datasets as follows: 1) broodstock spawners from brood years 2005 – 2011 with information on the number of offspring recaptured as adults per broodstock pair, and 2) adult fish caught by anglers during 20 run years over a 30-year period and after scale analysis being categorized as either wild-born or hatchery-reared. Individuals identified as farmed escapees based on scale analysis were removed."

Whilst escaped farmed salmon are generally known to be easier to catch on rod compared to wild salmon, there are no data suggesting that introgressed wild-born or hatchery-reared salmon are more easily caught on rod than wild-born or hatchery-reared salmon with no farmed ancestry. Nor is there any indication that hatchery-reared fish are more easily caught on rod than fish born in nature. Because all hatchery-reared fish compared in this study were released as smolts and would have spent at least one year at sea before capture as returning adults in the river, it is unlikely that different degrees of introgression should lead to different capture probabilities.

This article can be considered as a follow-up study of their previous paper reporting the phenotypic effect of introgression (Bolstad et al. 2017). However, I do not see any description that tried to relate the reproductive success with their phenotypes as potential mechanism of the reported phenomenon.

We thank the reviewer for this suggestion and we have conducted analysis on sea age accordingly. No effect of introgression of sea-age was found in our data, thus indicating that decreased sea-age does not contribute to increased survival of introgressed individuals. These results have been included in the Results section as follows (Lines 155-159): “Introgressed individuals are expected to spend fewer years at sea⁴⁰. In the Eira, we found no apparent effect of introgression on sea age (Supplementary Fig. 2; Supplementary Table 8), and a potential higher survival by spending shorter time at sea has therefore likely not contributed to a higher return rate of offspring from introgressed broodstock.”

Moreover, we have been able to relate individual introgression to the size of out-migrating smolts. The new analysis shows that introgressed hatchery-reared individuals were significantly larger as out-migrating smolts than hatchery-reared individuals with no farmed ancestry. This result sheds light on the biological mechanisms underlying the increased number of adult offspring produced by introgressed broodstock and indicate that introgressed juveniles grow larger in the hatchery and is therefore likely to have increased survival at sea. These results are presented under the sub-heading “Effect of introgression on growth in hatchery-reared individuals”.

Detailed comments

Abstract: no period in Line 16. More importantly, the study species should be declared earlier than the end of the abstract (Line 19).

In the new version of the Abstract, the study species is declared early.

Line 40: ‘escapees’ is an inappropriate wording for supplementation programs because they are meant to be released in the wild.

‘Escapees’ in this context refers to farmed individuals that have accidentally escaped and not intentionally released into the wild. The sentence is therefore left unchanged.

Line 227: Is Table S2 the same as the Supplementary Table 2 (GLMM parameters)?

More analyses have led to a new set of tables. We have taken care to ensure that tables are numbered correctly.

Line 239: How the phenotypic measurements, other than the fish origins, were used in this study is unclear.

Phenotypic measurements for broodstock included the weight of dams, which is used in the analysis. We agree with the reviewer that for adult spawners returning to the river, no other phenotypic measurements were used in the original analysis other than presence/absence of the adipose fin and scale analysis. The inclusion of total length was therefore redundant. In response to the suggestion of reviewer 3 we have included a new analysis in the revised version of the manuscript. This involves estimation of smolt size, which is estimated using scale growth patterns and total length at capture. As such, total length (mm; from the tip of the snout to the end of the caudal fin) remains as a phenotypic measurement for returning adult spawners.

Line 263: How were Pw and Pd determined, and what do they mean? For example, does “domesticated salmon” in this river have on average 90% of “domesticated genes” and 10% of “wild genes”? Detailed explanation is needed.

We appreciate that this description needs clarification. We have altered the description as follows (Lines 297-305): "Proportion of farmed ancestry (D) in each individual was determined from individual estimates of the probability of belonging to farmed salmon (P_ind) by scaling to the average estimates of probability of belonging to farmed salmon in a historical reference sample of pure wild salmon from the Eira (P_W = 0.0644) and to reference samples of farmed salmon from all breeding lines used in Norway (P_D = 0.903), according to the following formulae²:

$$D = (P_{\text{ind}} - P_W) / (P_D - P_W)$$

Note that the average P_D in Norwegian farm populations is less than one and the average P_W in the historical reference sample is above zero."

Reviewers' comments:

Reviewer #1 (Remarks to the Author):

I think the authors have generally responded well to the criticisms raised. However, reading through this new version of the paper there is one section that I find difficult to follow. This concerns the para from l. 101-140, where results and interpretation thereof become quite complex. For instance, it is not well explained why farmed ancestry has an effect on reproductive success of wild-born broodstock but not on hatchery-reared broodstock. You state that this is "surprising" and provide an explanation (for lack of effect of farmed ancestry in hatchery-reared broodstock) in l. 114-118. However, why should the same argument should not apply to wild-born broodstock? A clarification of results and their biological interpretation is warranted here. Also, if these results and their implications could be summed up in a few, clear sentences towards the end of the para it would be a great help to readers.

A few minor points:

l. 28-32. You provide references for nearly all of these negative effects, but not for "loss of adaptation", which is almost the most important. Please add a reference supporting this.

l.107 famed -> farmed

l. 133 I would prefer genotypes instead of genomes, as you have not studied genomes.

l. 348-349 This sentence does not read well.

Apart from this I can only say that I maintain my positive impression of this manuscript and find it to represent a really important, interesting and well-conducted study.

Reviewer #2 (Remarks to the Author):

Manuscript#: NCOMMS-18-11223A

Title: Supplementary stocking selects for domesticated genotypes

It has been a pleasure to read the revision of this manuscript. The authors have tried to reply to the comments made by myself and the other two reviewers in significant detail, and I think that most of the concerns could be addressed. The authors have requested additional data and performed new analyses in this effort, and I appreciated this. One thing that I think should be added to the main text of the study is a more detailed explanation on how they salmon were reared in the hatcheries. As it stands, it is unclear which hatchery procedures may have impacted salmon during their time in the hatcheries. I was concerned that size selection could have accounted for most of the fact that introgressed salmon were preferentially produced, but the authors have replied in their response that this is not the case. I think this needs to better come out in the text, since size selection is a standard procedure in the majority of hatcheries to remove stunted or poor performing fish.

Reviewer #3 (Remarks to the Author):

I think that the authors replied to the reviewers' comments appropriately, and that the manuscript is ready for publication.

Reviewers' comments:

Reviewer #1 (Remarks to the Author):

I think the authors have generally responded well to the criticisms raised. However, reading through this new version of the paper there is one section that I find difficult to follow. This concerns the para from l. 101-140, where results and interpretation thereof become quite complex. For instance, it is not well explained why farmed ancestry has an effect on reproductive success of wild-born broodstock but not on hatchery-reared broodstock. You state that this is "surprising" and provide an explanation (for lack of effect of farmed ancestry in hatchery-reared broodstock) in l. 114-118. However, why should the same argument should not apply to wild-born broodstock? A clarification of results and their biological interpretation is warranted here. Also, if these results and their implications could be summed up in a few, clear sentences towards the end of the para it would be a great help to readers.

We thank the reviewer for pointing out a mistake in the aforementioned Lines 114 – 118 (now lines 110-116). These lines lacked the term 'second generation'. Moreover, we have attempted to improve the biological explanation as follows: "While this lack of an effect of farmed ancestry in hatchery-reared broodstock is surprising, a possible explanation may be that a positive effect in the hatchery is counteracted after release by a larger negative effect of being second generation hatchery-reared. Multiple generations in captivity may cause cumulative negative effects on fitness components in salmonids⁴⁷. However, the combined effects of introgression and captive rearing and how these factors affect different life history stages of wild salmon is largely unknown."

Also, we agree that the paragraph benefited from clarification and we have therefore altered the order at which the results are presented within the paragraph. As requested, the main results and implications have been summed up in a sentence at the end of the paragraph as follows (Lines 138 – 143): "The environmental background of dams, egg number and weight of the dams (which affects egg number) influenced the number of recaptured offspring, but with smaller effect sizes than that of introgression (Table 1, Supplementary Tables 5 and 6). Controlling for these factors did not diminish the effect of introgression, which under hatchery conditions may lead to a more than four-fold increase in reproductive success for wild-born individuals."

A few minor points:

l. 28-32. You provide references for nearly all of these negative effects, but not for "loss of adaptation", which is almost the most important. Please add a reference supporting this.

We have added the following reference: Laikre L, Schwartz MK, Waples RS, Ryman N. Compromising genetic diversity in the wild: unmonitored large-scale release of plants and animals. Trends in Ecology & Evolution 25, 520-529 (2010).

l.107 famed -> farmed

The typo has been corrected.

l. 133 I would prefer genotypes instead of genomes, as you have not studied genomes.

We have changed the text accordingly.

l. 348-349 This sentence does not read well.

We have changed the text as follows: "The effect of introgression on sea age (measured as probability of maturing given survival to adulthood) was analysed using the following multinomial

(logit) mixed effect models:”.

Apart from this I can only say that I maintain my positive impression of this manuscript and find it to represent a really important, interesting and well-conducted study.

Reviewer #2 (Remarks to the Author):

Manuscript#: NCOMMS-18-11223A

Title: Supplementary stocking selects for domesticated genotypes

It has been a pleasure to read the revision of this manuscript. The authors have tried to reply to the comments made by myself and the other two reviewers in significant detail, and I think that most of the concerns could be addressed. The authors have requested additional data and performed new analyses in this effort, and I appreciated this. One thing that I think should be added to the main text of the study is a more detailed explanation on how they salmon were reared in the hatcheries. As it stands, it is unclear which hatchery procedures may have impacted salmon during their time in the hatcheries. I was concerned that size selection could have accounted for most of the fact that introgressed salmon were preferentially produced, but the authors have replied in their response that this is not the case. I think this needs to better come out in the text, since size selection is a standard procedure in the majority of hatcheries to remove stunted or poor performing fish.

We understand the reviewer’s concern. We have requested additional information from the hatchery managers and based on this information we added the following text to the Methods section (Lines 258 – 271): “The most significant mortality was observed prior to hatching and during the period of initial feeding. After the initial feeding, the mortality is reported to be zero to five individuals per month. The fish were kept at densities of 15 kg/m³ during initial feeding and 25 kg/m³ during growth. Such low densities may increase territoriality and aggressive behaviour⁷⁰. Sick or injured individuals were removed. Feed size and change to larger pellet feed was based on the average size of fish in each holding basin. All fish were sorted and moved to larger holding basins together with conspecifics of similar size at two separate stages: during their first summer (age 0+) and again during their second summer (age 1+). Keeping the fish with individuals of similar size minimizes competition and allows for similar growth rates. No deliberate culling of small or poor performing fish has occurred. The fish were moved to net pens at the outlet of Lake Eikesdalsvatnet - which is the source of River Eira - for acclimatisation prior to release into the river either during the second summer (one-year smolts) or third summer (two-year smolts).”

The selection pressure this handling is likely to have induced is elaborated upon in the Discussion as follows (Lines 195-204): “Given the difference in number of fertilised eggs and released smolts, there is a large potential for selection in the hatchery, particularly at the stage of initial feeding, when the highest mortality was observed. Selection in favour of introgressed individuals at the stage of initial feeding is expected, given that farmed⁶⁰ and hybrid⁶¹ individuals are known to outcompete wild salmon when held in sympatry at the early life history stage following emergence. Because introgressed hatchery-reared individuals were larger at release as two-year smolts whilst mortality in the hatchery was low during the growth phase, it is likely that introgressed individuals have been favoured at two distinct life history stages: first in the hatchery during initial feeding due to competitive behaviour⁶⁰ and faster growth²³ and then at sea, where a large size is expected to increase survival⁵³.”

Reviewer #3 (Remarks to the Author):

I think that the authors replied to the reviewers' comments appropriately, and that the manuscript is ready for publication.

REVIEWERS' COMMENTS:

Reviewer #1 (Remarks to the Author):

I appreciate the thorough revision undertaken in response to my comments, and I have no more issues to raise. This is a great paper and in my opinion it is now acceptable for publication.

Reviewer #2 (Remarks to the Author):

2nd review of 'Supplementary stocking selects for domesticated genotypes'.

I think the authors have done a good job to address all remaining comments and I have no further concerns.

REVIEWERS' COMMENTS:

Reviewer #1 (Remarks to the Author):

I appreciate the thorough revision undertaken in response to my comments, and I have no more issues to raise. This is a great paper and in my opinion it is now acceptable for publication.

Reviewer #2 (Remarks to the Author):

2nd review of 'Supplementary stocking selects for domesticated genotypes'.

I think the authors have done a good job to address all remaining comments and I have no further concerns.

Response

No changes were requested by the reviewers.